# Extracting Latent Structure From Multiple Interacting Neural Populations

**João D. Semedo**[1,2,3]**, Amin Zandvakili**[4]**, Adam Kohn**[4]**,**
*****Christian K. Machens**[3]**, ***Byron M. Yu**[1,5]
[1]Department of Electrical and Computer Engineering, Carnegie Mellon University
[2]Department of Electrical and Computer Engineering, Instituto Superior Técnico
[3]Champalimaud Neuroscience Programme, Champalimaud Center for the Unknown
[4]Dominick Purpura Department of Neuroscience, Albert Einstein College of Medicine
[5]Department of Biomedical Engineering, Carnegie Mellon University
jsemedo@cmu.edu      {amin.zandvakili,adam.kohn}@einstein.yu.edu
christian.machens@neuro.fchampalimaud.org     byronyu@cmu.edu
***** Denotes equal contribution.

## Abstract

Developments in neural recording technology are rapidly enabling the recording of populations of neurons in multiple brain areas simultaneously, as well as the identification of the types of neurons being recorded (e.g., excitatory vs. inhibitory). There is a growing need for statistical methods to study the interaction among multiple, labeled populations of neurons. Rather than attempting to identify direct interactions between neurons (where the number of interactions grows with the number of neurons squared), we propose to extract a smaller number of latent variables from each population and study how these latent variables interact. Specifically, we propose extensions to probabilistic canonical correlation analysis (pCCA) to capture the temporal structure of the latent variables, as well as to distinguish within-population dynamics from across-population interactions (termed Group Latent Auto-Regressive Analysis, gLARA). We then applied these methods to populations of neurons recorded simultaneously in visual areas V1 and V2, and found that gLARA provides a better description of the recordings than pCCA. This work provides a foundation for studying how multiple populations of neurons interact and how this interaction supports brain function.

## 1 Introduction

In recent years, developments in neural recording technologies have enabled the recording of populations of neurons from multiple brain areas simultaneously [1–7]. In addition, it is rapidly becoming possible to identify the types of neurons being recorded (e.g., excitatory versus inhibitory [8]). Enabled by these experimental advances, a major growing line of scientific inquiry is to ask how different populations of neurons interact, whether the populations correspond to different brain areas or different neuron types. To address such questions, we need statistical methods that are well-suited for assessing how different groups of neurons interact on a population level.

One way to characterize multi-population activity is to have the neurons interact directly [9–11], then examine the properties of the interaction strengths. While this may be a reasonable approach for small populations of neurons, the number of interactions grows with the square of the number of recorded neurons, which may make it difficult to summarize how larger populations of neurons interact [12]. Instead, it may be possible to obtain a more succinct account by extracting latent variables for each population and asking how these latent variables interact.

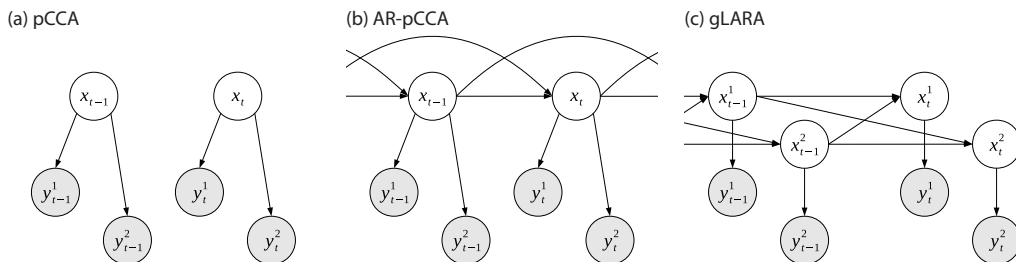

Figure 1: **Directed graphical models for multi-population activity.** **(a)** Probabilistic canonical correlation analysis (pCCA). **(b)** pCCA with auto-regressive latent dynamics (AR-pCCA). **(c)** Group latent auto-regressive analysis (gLARA). For clarity, we show only two populations in each panel and auto-regressive dynamics of order 1 in panel **(c)**.

Dimensionality reduction methods have been widely used to extract succinct representations of population activity [13–17] (see [18] for a review). Each observed dimension corresponds to the spike count (or firing rate) of a neuron, and the goal is to extract latent variables that describe how the population activity varies across experimental conditions, experimental trials, and/or across time. These previous studies use dimensionality reduction methods that do not explicitly account for multiple populations of neurons. In other words, these methods are invariant to permutations of the ordering of the neurons (i.e., the observed dimensions).

This work focuses on latent variable methods designed explicitly for studying the interaction between labelled populations of neurons. To motivate the need for these methods, consider applying a standard dimensionality reduction method, such as factor analysis (FA) [19], to all neurons together by ignoring the population labels. The extracted latent variables would capture all modes of covariability across the neurons, without distinguishing between-population interaction (i.e., the quantity of interest) from within-population interaction. Alternatively, one might first apply a standard dimensionality reduction method to each population of neurons individually, then examine how the latent variables extracted from each population interact. However, important features of the between-population interaction may be eliminated by the dimensionality reduction step, whose sole objective is to preserve the within-population interaction.

We begin by considering canonical correlations analysis (CCA) and its probabilistic formulation (pCCA) [20], which identify a single set of latent variables that explicitly captures the between-population covariability. To understand how the different neural populations interact on different timescales, we propose extensions of pCCA that introduce a separate set of latent variables for each neural population, as well as dynamics on the latent variables to describe their interaction over time. We then apply the proposed methods to populations of neurons recorded simultaneously in visual areas V1 and V2 to demonstrate their utility.

## 2  Methods

We consider the setting where many neurons are recorded simultaneously, and the neurons belong to distinct populations (either by brain area or by neuron type). Let $\mathbf{y}_t^i \in \mathbb{R}^{q_i}$ represent the observed activity vector of population $i \in \{1, ..., M\}$ at time $t \in \{1, ..., T\}$, where $q_i$ denotes the number of neurons in population $i$. Below, we consider three different ways to study the interaction between the neural populations. To keep the notation simple, we'll only consider two populations ($M = 2$); the extension to more than two populations is straightforward.

### 2.1  Factor analysis and probabilistic canonical correlation analysis

Consider the following latent variable model, that defines a linear-Gaussian relationship between the observed variables, $\mathbf{y}_t^1$ and $\mathbf{y}_t^2$, and the latent state, $\mathbf{x}_t \in \mathbb{R}^p$:

$$\mathbf{x}_t \sim \mathcal{N}\left(\mathbf{0}, I\right) \tag{1}$$

$$\begin{bmatrix} \mathbf{y}_t^1 \\ \mathbf{y}_t^2 \end{bmatrix} \mid \mathbf{x}_t \sim \mathcal{N}\left( \begin{bmatrix} C^1 \\ C^2 \end{bmatrix} \mathbf{x}_t + \begin{bmatrix} \mathbf{d}^1 \\ \mathbf{d}^2 \end{bmatrix}, \begin{bmatrix} R^{11} & R^{12} \\ R^{12^T} & R^{22} \end{bmatrix} \right) \tag{2}$$

where $C^i \in \mathbb{R}^{q_i \times p}$, $\mathbf{d}^i \in \mathbb{R}^{q_i}$ and:

$$\begin{bmatrix} R^{11} & R^{12} \\ R^{12^T} & R^{22} \end{bmatrix} \in \mathbb{S}_{++}^q$$

with $q = q_1 + q_2$. According to this model, the covariance of the observed variables is given by:

$$\mathrm{cov}\left( \begin{bmatrix} \mathbf{y}_t^1 \\ \mathbf{y}_t^2 \end{bmatrix} \right) = \begin{bmatrix} C^1 \\ C^2 \end{bmatrix} \begin{bmatrix} C^{1^T} & C^{2^T} \end{bmatrix} + \begin{bmatrix} R^{11} & R^{12} \\ R^{12^T} & R^{22} \end{bmatrix} \tag{3}$$

Factor analysis (FA) and probabilistic canonical correlation analysis (pCCA) can be seen as two special cases of the general model presented above. FA assumes the noise covariance to be diagonal, i.e., $R^{11} = \mathrm{diag}(r_1^1, ..., r_{q_1}^1)$, $R^{22} = \mathrm{diag}(r_1^2, ..., r_{q_2}^2)$ and $R^{12} = \mathbf{0}$. This noise covariance captures only the independent variance of each neuron, and not the covariance between neurons. As a result, the covariance between neurons is explained by the latent state through the observation matrices $C^1$ and $C^2$. pCCA, on the other hand, considers a block diagonal noise covariance, i.e., $R^{12} = \mathbf{0}$. This noise covariance accounts for the covariance observed between neurons in the same population. The latent state is therefore only used to explain the covariance between neurons in different populations. The directed graphical model for pCCA is shown in Fig.1a.

## 2.2 Auto-regressive probabilistic canonical correlation analysis (AR-pCCA)

While pCCA offers a succinct picture of the covariance structure between populations of neurons, it does not capture any temporal structure. There are two main reasons as to why this time structure may be interesting. First, pCCA is modelling the covariance structure at zero time lag, which may not capture all of the interactions of interest. If the two populations of neurons correspond to two different brain areas, there may be important interactions at non-zero time lags due to physical delays in information transmission. Second, the two populations of neurons may interact at more than one time delay, for example if multiple pathways exist between the neurons in these populations. To take the temporal structure into account we will first extend pCCA by defining an auto-regressive linear-Gaussian model on the latent state:

$$\mathbf{x}_t \sim \mathcal{N}\left( \mathbf{0}, I \right), \quad \text{if } 1 \leq t \leq \tau \tag{4}$$

$$\mathbf{x}_t \mid \mathbf{x}_{t-1}, \mathbf{x}_{t-2}, ..., \mathbf{x}_{t-\tau} \sim \mathcal{N}\left( \sum_{k=1}^{\tau} A_k \mathbf{x}_{t-k}, Q \right), \quad \text{if } t > \tau \tag{5}$$

where $A_k \in \mathbb{R}^{p \times p}, \forall k$, $Q \in \mathbb{S}_{++}^p$ and $\tau$ denotes the order of the autoregressive model. We term this model AR-pCCA, which is defined by the state model in Eq.(4)-(5) and the observation model in Eq.(2) with $R^{12} = \mathbf{0}$. Although the observation model is the same as that for pCCA, the latent state here accounts for temporal dynamics, as well as the covariation structure between the populations. The corresponding directed graphical model is shown in Fig.1b.

## 2.3 Group latent auto-regressive analysis (gLARA)

According to AR-pCCA, a single latent state drives the observed activity in both areas. As a result, it's not possible to distinguish the within-population dynamics from the between-population interactions. To allow for this, we propose using two separate latent states, one per population, that interact over time. We refer to the proposed model as group latent auto-regressive analysis (gLARA):

$$\mathbf{x}_t \sim \mathcal{N}\left( \mathbf{0}, I \right), \quad \text{if } 1 \leq t \leq \tau \tag{6}$$

$$\mathbf{x}_t^i \mid \mathbf{x}_{t-1}, \mathbf{x}_{t-2}, ..., \mathbf{x}_{t-\tau} \sim \mathcal{N}\left( \sum_{j=1}^{2} \sum_{k=1}^{\tau} A_k^{ij} \mathbf{x}_{t-k}^j, Q^i \right), \quad \text{if } t > \tau \tag{7}$$

$$\begin{bmatrix} \mathbf{y}_t^1 \\ \mathbf{y}_t^2 \end{bmatrix} \mid \mathbf{x}_t \sim \mathcal{N}\left( \begin{bmatrix} C^1 & \mathbf{0} \\ \mathbf{0} & C^2 \end{bmatrix} \begin{bmatrix} \mathbf{x}_t^1 \\ \mathbf{x}_t^2 \end{bmatrix} + \begin{bmatrix} \mathbf{d}^1 \\ \mathbf{d}^2 \end{bmatrix}, \begin{bmatrix} R^1 & \mathbf{0} \\ \mathbf{0} & R^2 \end{bmatrix} \right) \tag{8}$$

where $\mathbf{x}_t$ is obtained by stacking $\mathbf{x}_t^1 \in \mathbb{R}^{p_1}$ and $\mathbf{x}_t^2 \in \mathbb{R}^{p_2}$, the latent states for each population, $C^i \in \mathbb{R}^{q_i \times p_i}$, $A_k^{ij} \in \mathbb{R}^{p_i \times p_j}$ and $Q^i \in \mathbb{S}_{++}^{p_i}$, $\forall k$ and $i \in \{1, 2\}$. Note that the covariance structure observed on a population level now has to be completely reflected by the latent states (there are no shared latent variables in this model) and is therefore defined by the dynamics matrices $A_k^{ij}$, allowing for the separation of the within-population dynamics ($A_k^{11}$ and $A_k^{22}$) and the between-population interactions ($A_k^{12}$ and $A_k^{21}$). Furthermore, the interaction between the populations is asymmetrically defined by $A_k^{12}$ and $A_k^{21}$, allowing for a more in depth study of the way in each the two areas interact by comparing these across the various time delays considered. Note that gLARA represents a special case of the AR-pCCA model.

## 2.4 Parameter estimation for gLARA

The parameters of gLARA can be fit to the training data using the expectation-maximization (EM) algorithm. To do so, we start by defining the augmented latent state $\bar{\mathbf{x}}_t \in \mathbb{R}^{p\tau}$, with $p = p_1 + p_2$:

$$\bar{\mathbf{x}}_t = \begin{bmatrix} \bar{\mathbf{x}}_t^1 \\ \bar{\mathbf{x}}_t^2 \end{bmatrix} = \begin{bmatrix} \mathbf{x}_t^{1T} \dots & \mathbf{x}_{t-\tau}^{1}{}^{T} & \mathbf{x}_t^{2T} \dots & \mathbf{x}_{t-\tau}^{2}{}^{T} \end{bmatrix}^T \tag{9}$$

and the augmented observation vector $\bar{\mathbf{y}}_t \in \mathbb{R}^q$, with $q = q_1 + q_2$:

$$\bar{\mathbf{y}}_t = \begin{bmatrix} \mathbf{y}_t^{1T} & \mathbf{y}_t^{2T} \end{bmatrix}^T \tag{10}$$

for $t \in \{\tau, ..., T\}$. Using the augmented latent state $\bar{\mathbf{x}}$, the dynamics equation (Eq.(6) and (7)) can be rewritten as:

$$\bar{\mathbf{x}}_t \sim \mathcal{N}(\mathbf{0}, I), \quad \text{if } t = \tau \tag{11}$$

$$\bar{\mathbf{x}}_t \mid \bar{\mathbf{x}}_{t-1} \sim \mathcal{N}(\bar{A}\bar{\mathbf{x}}_{t-1}, \bar{Q}), \quad \text{if } t > \tau \tag{12}$$

for appropriately structured $\bar{A} \in \mathbb{R}^{p\tau \times p\tau}$ and $\bar{Q} \in \mathbb{S}_{++}^{p\tau}$. The observation model (Eq.(8)) can be rewritten as:

$$\bar{\mathbf{y}}_t \mid \bar{\mathbf{x}}_t \sim \mathcal{N}\left( \bar{C} \begin{bmatrix} \bar{\mathbf{x}}_t \\ 1 \end{bmatrix}, \bar{R} \right) \tag{13}$$

for appropriately structured $\bar{C} \in \mathbb{R}^{q \times (p\tau+1)}$ and $\bar{R} \in \mathbb{S}_{++}^{q}$. Due to space constraints, we will not explicitly show the structure of the augmented parameters $\bar{\theta} = \{\bar{C}, \bar{R}, \bar{A}, \bar{Q}\}$. It is straightforward to derive them by inspection of Eq.(9)-(13).

We fit the model parameters using the EM algorithm. In the E-step, because the latent and observed variables are jointly Gaussian, $P(\bar{\mathbf{x}}_t \mid \bar{\mathbf{y}}_1, ..., \bar{\mathbf{y}}_T)$ is also Gaussian and can be computed exactly by applying the forward-backward recursion of the Kalman smoother [21] on the augmented vectors. In the M-step, we directly estimate the original parameters $\theta = \{C^i, \mathbf{d}^i, R^i, A_k^{ij}\}$, as opposed to estimating the structured form of the augmented parameters $\bar{\theta} = \{\bar{C}, \bar{R}, \bar{A}\}$ (without loss of generality, we set $Q^i = I$):

$$\begin{bmatrix} C^i & \mathbf{d}^i \end{bmatrix} = \left( \sum_{t=1}^{T} \mathbf{y}_t^i \begin{bmatrix} \mathbb{E}(\mathbf{x}_t^{iT}) & 1 \end{bmatrix} \right) \left( \sum_{t=1}^{T} \begin{bmatrix} \mathbb{E}(\mathbf{x}_t^i \mathbf{x}_t^{iT}) & \mathbb{E}(\mathbf{x}_t^i) \\ \mathbb{E}(\mathbf{x}_t^{iT}) & 1 \end{bmatrix} \right)^{-1} \tag{14}$$

$$R^i = \frac{1}{T} \sum_{t=1}^{T} \{ (\mathbf{y}_t^i - \mathbf{d}^i)(\mathbf{y}_t^i{}^T - \mathbf{d}^i) - C^i \mathbb{E}(\mathbf{x}_t^i)(\mathbf{y}_t^i - \mathbf{d}^i)^T$$

$$- (\mathbf{y}_t^i - \mathbf{d}^i) \mathbb{E}(\mathbf{x}_t^{iT}) C^{iT} + C^i \mathbb{E}(\mathbf{x}_t^i \mathbf{x}_t^{iT}) C^{iT} \} \tag{15}$$

$$\begin{bmatrix} A_1^{11} & \dots & A_k^{11} & A_1^{12} & \dots & A_k^{12} \\ A_1^{21} & \dots & A_k^{21} & A_1^{22} & \dots & A_k^{22} \end{bmatrix} = \left( \sum_{t=2}^{T} \mathbb{E}(\bar{\mathbf{x}}_t \bar{\mathbf{x}}_{t-1}^T) \right) \left( \sum_{t=2}^{T} \mathbb{E}(\bar{\mathbf{x}}_{t-1} \bar{\mathbf{x}}_{t-1}^T) \right)^{-1} \tag{16}$$

To initialize the EM algorithm, we start by applying FA to each population individually, and use the estimated observation matrices $C^1$ and $C^2$, as well as the mean vectors $\mathbf{d}^1$ and $\mathbf{d}^2$ and the observation covariance matrices $R^{11}$ and $R^{22}$. The $A_k^{ij}$ matrices are initialized at $\mathbf{0}$.

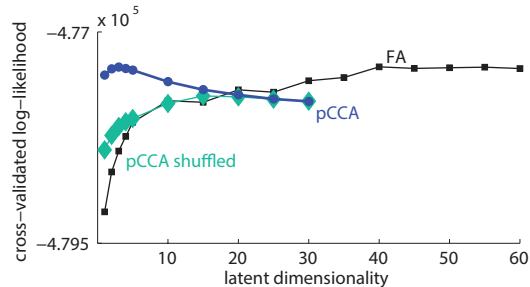

Figure 2: **Comparing the optimal dimensionality for FA and pCCA. (a)** Cross-validated log-likelihood plotted as a function of the dimensionality of the latent state for FA (black) and pCCA (blue). pCCA was also applied to the same data after randomly shuffling the population labels (green). Note that maximum possible dimensionality for pCCA is 31, which is the size of the smaller of the two populations (in this case, V2).

## 2.5  Neural recordings

The methods described above were applied to multi-electrode recordings performed simultaneously in visual area 1 (V1) and visual area 2 (V2) of an anaesthetised monkey, while the monkey was shown a set of oriented gratings with 8 different orientations. Each of the 8 orientations was shown 400 times for a period of 1.28s, providing a total of 3200 trials. We used 1.23s of data in each trial, from 50ms after stimulus onset until the end of the trial, and proceeded to bin the observed spikes with a 5ms window. The recordings include a total of 97 units in V1 and 31 units in V2 (single- and multi-units). For model comparison, we performed 4-fold cross-validation, splitting the data into four non-overlapping test folds with 250 trials each. We chose to analyze a subset of the trials for rapid iteration of the analyses, as the cross-validation procedure is computationally expensive for the full dataset. Given that 1000 trials provides a total of 246,000 timepoints (at 5 ms resolution), this provides a reasonable amount of data to fit any of the models with the 128 observed neurons.

In this study, we sought to investigate how trial-to-trial population variability in V1 relates to the trial-to-trial population variability in V2. For these gratings stimuli (which are relatively simple compared to naturalistic stimuli [22]), there is likely richer structure in the V1-V2 interaction for the trial-to-trial variability than for the stimulus drive. To this end, we preprocessed the neural activity by computing the peristimulus time histogram (PSTH), representing the trial-averaged firing rate timecourse, for each neuron and experimental condition (grating orientation). For each spike train, we then subtracted the appropriate PSTH from the binned spike counts to obtain a single-trial "residual". The residuals across all neurons and conditions were considered together in the analyses shown in Section 3. Note that the methods considered in this study could also be applied to the PSTHs of sequentially recorded neurons in multiple areas.

## 3  Results

We started by asking how many dimensions are needed to describe the between-population covariance, relative to the number of dimensions needed to describe the within-population covariance. This was assessed by applying pCCA to the labeled V1 and V2 populations, as well as FA to the two populations together (which ignores the V1 and V2 labels). In this analysis, pCCA captures only the between-population covariance, whereas FA captures both the between-population and within-population covariance. By comparing cross-validated data likelihoods for different dimensionalities, we found that pCCA required three latent dimensions, whereas FA required 40 latent dimensions (Fig.2). This indicates that the zero time lag interaction between V1 and V2 is confined to a small number of dimensions (three) relative to the number of dimensions (40) needed to describe all co-variance among the neurons. The difference of these two dimensionalities (37) describes covariance that is 'private' to each population (i.e., within-population covariance). The FA and pCCA curves peak at similar cross-validated likelihoods in Fig.2 because the observation model for pCCA Eq.(2) accounts for the within-population covariance (which is not captured by the pCCA latents).

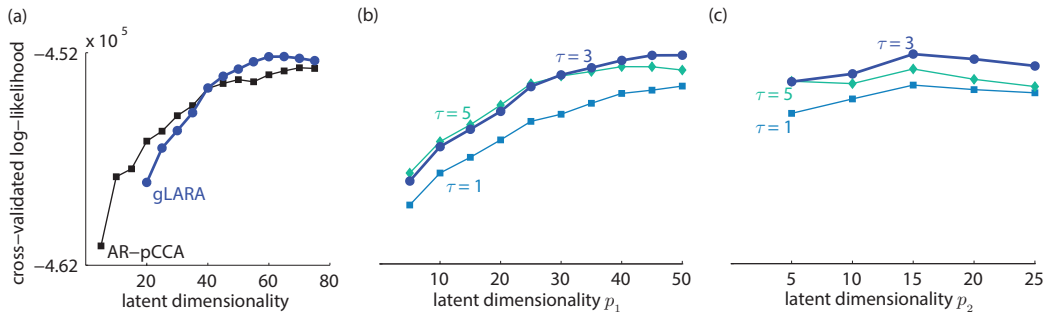

Figure 3: **Model selection for AR-pCCA and gLARA.** **(a)** Comparing AR-pCCA and gLARA as a function of the latent dimensionality (defined as $p_1 + p_2$ for gLARA, where $p_2$ was fixed at 15), for $\tau = 3$. **(b)** gLARA's cross-validated log-likelihood plotted as a function of the dimensionality of V1's latent state, $p_1$ (for $p_2 = 15$), for different choices of $\tau$. **(c)** gLARA's cross-validated log-likelihood plotted as a function of the dimensionality of V2's latent state, $p_2$ (for $p_1 = 50$), for different choices of $\tau$.

The distinction between within-population covariance and between-population covariance is further supported by re-applying pCCA, but now randomly shuffling the population labels. The cross-validated log-likelihood curve for these mixed populations now peaks at a larger dimensionality than three. The reason is that the shuffling procedure removes the distinction between the two types of covariance, such that the pCCA latents now capture both types of covariance (of the original unmixed populations). The peak for mixed pCCA occurs at a lower dimensionality than for FA for two reasons: i) because the mixed populations have the same number of neurons as the original populations (97 and 31), the maximum number of dimensions that can be identified by pCCA is 31, and ii) for the same latent dimensionality, pCCA has a larger number of parameters than FA, which makes pCCA more prone to overfitting.

Together, the analyses in Fig.2 demonstrate two key points. First, if the focus of the analysis lies in the interaction between populations, then pCCA provides a more parsimonious description, as it focuses exclusively on the covariance between populations. In contrast, FA is unable to distinguish within-population covariance from between-population covariance. Second, the neuron groupings for V1 and V2 are meaningful, as the number of dimensions needed to describe the covariance between V1 and V2 is small relative to that within each population.

We then analysed the performance of the models with latent dynamics (AR-pCCA and gLARA). The cross-validated log-likelihood for these models depends jointly on the dimensionality of the latent state, $p$, and the order of the auto-regressive model, $\tau$. For gLARA, $p$ is the sum of the dimensionalities of each population's latent state, $p_1 + p_2$, and we therefore want to jointly maximize the cross-validated log-likelihood with respect to both $p_1$ and $p_2$. AR-pCCA required a latent dimensionality of $p = 70$, while gLARA peaked for a joint latent dimensionality of 65 ($p_1 = 50$ and $p_2 = 15$) (Fig.3a). When computing the performance of AR-pCCA we considered models with $p \in \{5, 10, ..., 75\}$ and $\tau \in \{1, 3, ..., 7\}$ (Fig.3a shows the $\tau = 3$ case). To access how gLARA's cross-validated log-likelihood varied with the latent dimensionalities and the model order, we plotted it in Fig.3b, for $p_2 = 15$ and $p_1 \in \{5, 10, ..., 50\}$, for different choices of $\tau$. This showed that the performance is greater for an order 3 model, and that it saturates by the time $p_1$ reaches 50. In Fig.3c, we did a similar analysis for the dimensionality of V2's latent state, where $p_1$ was held constant at 50 and $p_2 \in \{5, 10, ..., 25\}$. The cross-validated log-likelihood shows a clear peak at $p_2 = 15$ regardless of $\tau$. We found that, for both models, the cross-validated log-likelihood peaks for $\tau = 3$ (see Fig.3b and 3c for gLARA, results not shown for AR-pCCA).

Finally, we asked which model, AR-pCCA or gLARA, better describes the data. Note that gLARA is a special case of AR-pCCA, where the observation matrix in Eq.(8) is constrained to have a block diagonal structure (with blocks $C^1$ and $C^2$). The key difference between the two models is that gLARA assigns a non-overlapping set of latent variables to each population. We found that gLARA outperforms AR-pCCA (Fig.3a). This suggests that the extra flexibility of the AR-pCCA model

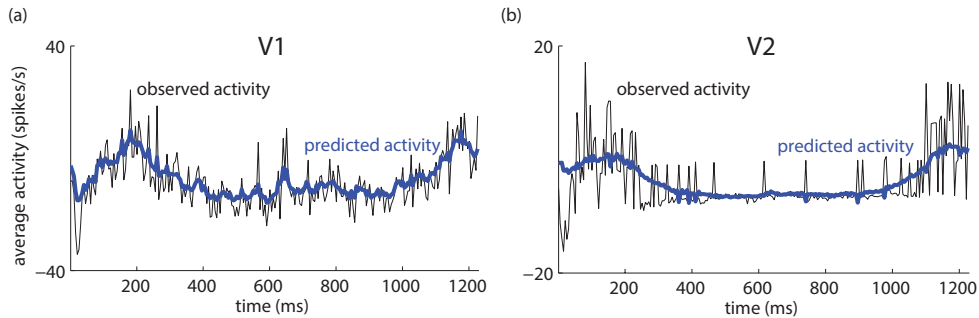

Figure 4: **Leave-one-neuron-out prediction using gLARA.** Observed activity (black) and the leave-one-neuron-out prediction of gLARA (blue) for a representative held-out trial, averaged over **(a)** the V1 population and **(b)** the V2 population. Note that the activity can be negative because we are analyzing the single-trial residuals (cf. Section 2.5).

leads to overfitting and that the data are better explained by considering two separate sets of latent variables that interact.

The optimal latent dimensionalities found for AR-pCCA and gLARA are substantially higher than those found for pCCA, as the latent states now also capture non-zero time lag interactions between the populations, and the dynamics within each population. For gLARA, the between-population covariance must be accounted for by the interaction between the population-specific latents, $\mathbf{x}_t^1$ and $\mathbf{x}_t^2$, because there are no shared latents in this model. Thus, the interaction between V1 and V2 is summarized by the $A_k^{12}$ and $A_k^{21}$ matrices. Also, both AR-pCCA and gLARA outperform FA and pCCA (comparing vertical axes in Fig.2 and 3), showing that there is meaningful temporal structure in how V1 and V2 interact that can be captured by these models.

Having performed a systematic, relative comparison between AR-pCCA and gLARA models of different complexities, we asked how well the best gLARA model fit the data in an absolute sense. To do so, we used $3/4$ of the data to fit the model parameters and performed leave-one-neuron-out prediction [15] on the remaining $1/4$. This is done by estimating the latent states $\mathbb{E}\left(\mathbf{x}_{1,...,T}^1 \mid \mathbf{y}_{1,...,T}^1\right)$ and $\mathbb{E}\left(\mathbf{x}_{1,...,T}^2 \mid \mathbf{y}_{1,...,T}^2\right)$ using all but one neuron. This estimate of the latent state is then used to predict the activity of the neuron that was left out (the same procedure was repeated for each neuron). For visualization purposes, we averaged the predicted activity across neurons for a given trial and compared it to the recorded activity averaged across neurons for the same trial. We found that they indeed tracked each other, as shown in Fig.4 for a representative trial.

Finally, we asked whether gLARA reveals differences in the time structures of the within-population dynamics and the between-population interactions. We computed the Frobenius norm of both the within-population dynamics matrices $A_k^{11}$ and $A_k^{22}$ (Fig.5a) and the between-population interaction matrices $A_k^{12}$ and $A_k^{21}$ (Fig.5b), for $p_1 = 50$, $p_2 = 15$ and $\tau = 3$ ($k \in \{1, 2, 3\}$), which is the model for which the cross-validated log-likelihood was the highest. The time structure of the within-population dynamics appears to differ from that of the between-population interaction. In particular, the latents for each area depend more strongly on its own previous latents as the time delay increases up to 15 ms (Fig.5a). In contrast, the dependence between areas is stronger at time lags of 5 and 15 ms, compared to 10 ms (Fig.5b). Note that the peak of the cross-validated log-likelihood for $\tau = 3$ (Fig.3) shows that delays longer than 15ms do not contribute to an increase in the accuracy of the model and, therefore, the most significant interactions between these areas may occur within this time window. The structure seen in Fig.5 is not present if the same analysis is performed on data that are shuffled across time (results not shown). Because the latent states may have different scales, it is not informative to compare the magnitude of $A_k^{12}$ and $A_k^{21}$ or $A_k^{11}$ and $A_k^{22}$ ($A_k^{11}$ and $A_k^{22}$ also have different dimensions). Thus, we divided the norms for each $A_k^{ij}$ matrix by the respective maximum across $k$.

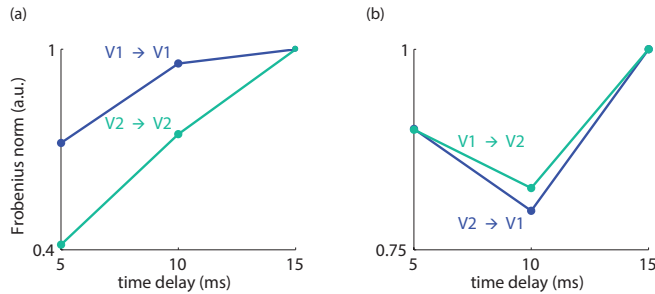

Figure 5: **Temporal structure of coupling matrices for gLARA.** **(a)** Frobenius norm of the within-population dynamics matrices $A_k^{11}$ and $A_k^{22}$, for $k \in \{1, 2, 3\}$. Each curve was divided by its maximum value. **(b)** Same as **(a)** for the between-population interaction matrices $A_k^{12}$ and $A_k^{21}$.

## 4   Discussion

We started by applying standard methods, FA and pCCA, to neural activity recorded simultaneously from visual areas V1 and V2. We found that the neuron groupings by brain area are meaningful, as the covariance of the neurons across areas is lower dimensional than that within each area. We then proposed an extension to pCCA that takes temporal dynamics into account and allows for the separation of within-population dynamics from between-population interactions (gLARA). This method was then shown to provide a better characterization of the two-population neural activity than FA and pCCA.

In the context of studying the interaction between populations of neurons, capturing the information flow is key to understanding how information is processed in the brain [3–7, 23]. To do so, one must be able to characterize the directionality of these between-population interactions. Previous studies have sought to identify the directionality of interactions directly between neurons, using measures such as Granger causality [10] (and related extensions, such as directed transfer function (DTF) [24]), and directed information [11]. Here, we proposed to study between-population interaction on the level of latent variables, rather than of the neurons themselves. The advantage is that this approach scales better with the number of recorded neurons and provides a more succinct picture of the structure of these interactions. To detect fine timescale interactions, it may be necessary to replace the linear-Gaussian model with a point process model on the spike trains [25].

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
