[Reviews · NeurIPS 2014]

Submitted by Assigned_Reviewer_16

It is becoming feasible to obtain cellular recordings with large numbers of cells simultaneously. In particular, this submission focuses on analysis when multiple populations of cells are recorded simultaneously (population being perhaps multiple brain areas, or cell types). In particular, they build latent variable models to describe the interactions between two, simultaneously recorded, neuronal populations.

The baseline method for modelling interactions between populations with latent variables is pCCA. However, CCA only captures instantaneous interactions, and does not capture any temporal structure.

The submission introduces AR-CCA, which defines a autoregressive linear-Guassian model on the latent state. In addition, they introduce gLARA which constrains the covariance structure by removing shared latent variables. They describe an expectation-maximization approach to fitting the model.

The authors then examine the efficacy of the models for describing a neuronal recording of V1 and V2 (actually the residuals after subtracting the PSTH). First, they demonstrated that pCCA, which distinguishes between within-population and between-population covariances is able to capture the data with many fewer latent variables than factor analysis (which does not distinguish).

Using cross-validation, they demonstrate that (unsurprisingly) AR-pCCA and gLARA, which contain dynamics are better able to model the data. gLARA is slightly better, which would indicate that the additional constraints in the model are helping to prevent overfitting. Finally, they examine the temporal structure of interactions between V1 and V2.

Quality
The claims made are well-supported by the evidence. The choice of pCCA is a reasonable baseline for comparison.

Clarity
The submission is well-written. In particular, the motivations behinds models and relationships between them is explained well which is very helpful for comprehension.

Originality

The manuscript introduces a new model AR-CCA which combines CCA with dynamics to allow capture relationships in sequential data. This is original, and the authors demonstrate its efficacy (along with the more constrained gLARA) in capturing neuronal interactions.

Significance
The submission is of primary interest to a specialised audience. While it is true that high-dimensional neuronal recordings are becoming increasingly common, the models presented are only relevant when multiple populations (known to the experimenter) are recorded simultaneously. Additionally, while the authors showed their models capture neuronal statistics (i.e. did well in cross-validation) it wasn’t clear what biological insights such models conveyed (for example, what does figure 5 help us understand about V1/V2 function). It would be helpful to at least speculate on this topic.
Summary: Well-written submission introducing new models for the analysis neuronal data from multiple populations. The models are somewhat specialised and not necessarily of general interest, the authors don’t convey any biological insights.

Submitted by Assigned_Reviewer_19

Summary:

The authors present a statistical method (gLARA) that captures dynamics within and between separate neural populations. They show parameter estimation using EM, compare the model to pCCA and then auto-regressive pCCA, and finally apply the method to a V1/V2 dataset.

Using pCCA and FA, they show that within-population variance requires several more factors or dimensions than needed to capture between-population variance. This result is for zero time lag. They then consider the dynamic models and show that gLARA provides superior cross-validation performance than AR-pCCA, but that both dynamic models capture meaningful between-population structure compared to pCCA/FA.

Quality:

gLARA is a simple, elegant, and principled formulation for tackling the problem at hand -- i.e. capturing within- and between-population dynamics with latent variables. The parameter estimation procedure is straightforward. The main results outlined above are interesting and convincing.

The authors carefully analyzed the models in terms of cross-validated performance. This is indeed a strong point of the paper. There is not as much discussion/analyses in terms of interpreting the identified gLARA model. As such, it is less clear how this method can aid questions about the nature of the within- and between-population dynamics themselves. The authors did show the Frobenius norms of A_1 to A_3, which show interesting patterns (e.g. the non-monotonicity of fig 5b). But since the norms max out at the max delay (15ms), are short (third order) FIR filters really an appropriate model for this data? Presumably so since tau=3 achieved superior cross-validated performance, but the shapes in fig 5 still seem unexpected.

Additionally, the dataset is particularly simple -- static presentations of oriented gratings. I think the paper would benefit from a discussion of how gLARA might be extended (if necessary) to handle richer datasets. For example, this paper focuses only on trial-by-trial dynamics, but what about data with richer input dynamics -- would it be necessary to extend to an ARX/ARMAX framework in order to truly differentiate between- and within-population dynamics?

Clarity:

The writing is exceptionally clear. A strong point is how clearly the authors explain the different models -- their similarities and differences in terms of covariance structure (sections 2.1-2.3).

Originality:

This paper is an original contribution.

Significance:

The problem that this paper addresses is of significant importance to the neuroscience community, particularly as datasets grow in size. Thus, this particular approach has potential to see widespread use.
Summary: This paper addresses an important problem and offers a simple framework for addressing within- and between-population neural dynamics. The paper is well-written and analyses are thorough.

Submitted by Assigned_Reviewer_42

This paper extends canonical correlation analysis of neural population recordings to the temporal domain, by having the latents interact with each other in an autoregressive manner. Moreover, the emphasis is on analysing data in which each neuron is labeled as belongings to one of several distinct populations. To tease apart within-population and between-population interactions, the latent space is explicitly divided into separate sets of latent variables, one for each population.

I found this paper very interesting and worth presenting at NIPS. The model is solid, very well motivated, explained, and dissected. The application to joint V1-V2 population recordings is nice too, and all explanations thereof made a lot of sense. The paper is well written.

The method will likely find applications in the neuroscience community.

I have a couple of minor issues:

- neural data (spikes binned in 5ms windows here) is very much non-Gaussian. Doesn't that pose some problems given the noise model considered here?

- The authors didn't show any equations for inference; in fact, they barely mentioned inference at all (except perhaps in the leave-one-out experiment: "This is done by estimating the latent states using all but one neuron"). I know inference is easy under a linear-Gaussian latent variable model, but it would be good to say a word or two about it.

- In the V1-V2 analysis, the extracted latent space was too big to be visualized. Do the authors think their method in general will find applications in dimensionality reduction / visualization of trial-to-trial variability?

- Computational considerations? Out of the 3200 available trials, only 1000 were used according to the authors. This is rather unusual practice, so I suspect the dataset was down-sampled to ease the computational burden? What dataset sizes do the authors think can be handled by their model?

Typos:
- 8 orientations x 400 trials = 3200 trials in total, not 3600 :)
- "the the" on p.6 (middle)

Summary: This paper formulates a novel generative model for neural recordings to study the interactions between and within distinct populations of neurons recorded simultaneously. I enjoyed reading it, and particularly liked the comparison to factor analysis and probabilistic canonical correlation analysis. Great work.
Author Feedback
Author rebuttal: We would like to thank the reviewers for all the helpful comments.

Assigned_Reviewer_16
"While it is true that high-dimensional neuronal recordings are becoming increasingly common, the models presented are only relevant when multiple populations (known to the experimenter) are recorded simultaneously."

While it is true that the models presented here require the labeling of different populations of neurons, such datasets are already commonly available and should became even more important in the future, with the continued development of electrical recording and optical imaging technologies. Some examples include Vazquez et al. (2013, PNAS), Saalmann et al. (2012, Science), Salazar et al. (2012, Science), Colgin et al. (2009, Nature) and Gregoriou et al. (2009, Science). Furthermore, the proposed methods are not restricted to electrical recordings and can be applied, for example, to calcium imaging datasets, which provide labelled neurons recorded simultaneously from different brain areas. Examples include the datasets obtained by Ahrens et al. (2013, Nature Methods) and Portugues et al. (2014, Neuron). Also, if the objective is to study trial-averaged dynamics, the proposed methods can be applied to the PSTHs of labelled neurons that are not recorded simultaneously. We intend to discuss both of these points in the final version of the paper.

Assigned_Reviewer_16
"Additionally, while the authors showed their models capture neuronal statistics (i.e. did well in cross- validation) it wasn’t clear what biological insights such models conveyed (for example, what does figure 5 help us understand about V1/V2 function)"

Fig.5 may point to the presence of two timescales of communication between these areas, which could reflect either feedforward and feedback paths, or two pathways of feedforward communication between V1 and V2. As suggested by the reviewer, we will speculate about this in the final paper. Future work will seek to reveal further insights about the interaction between the two brain areas.

Assigned_Reviewer_19
"(...) the dataset is particularly simple -- static presentations of oriented gratings. I think the paper would benefit from a discussion of how gLARA might be extended (if necessary) to handle richer datasets. For example, this paper focuses only on trial-by-trial dynamics, but what about data with richer input dynamics -- would it be necessary to extend to an ARX/ARMAX framework in order to truly differentiate between- and within-population dynamics?"

If one seeks to study trial-to-trial variability in the response to richer stimuli, we can take the same approach of subtracting the PSTH from each spike train and applying gLARA to the residuals. If one seeks to study the stimulus-driven response (i.e., the PSTHs) to either oriented gratings or richer stimuli, then we may need to extend gLARA to an ARX/ARMAX framework to account for the stimulus drive. We will discuss this in the final version of the paper.

Assigned_Reviewer_42
"Neural data (spikes binned in 5ms windows here) is very much non-Gaussian. Doesn't that pose some problems given the noise model considered here?"

The use of the Gaussian noise model has to do with the mathematical tractability of the fitting procedure. One can think of this noise model as capturing the first and second order statistics of the data. Fig.4 attempts to get at this by showing that this model is indeed able to capture the temporal structure in the two populations of neurons. In the future, we're considering substituting the Gaussian noise model by a point process model on the spike trains, as suggested in the paper, to deal with even smaller bin widths.

Assigned_Reviewer_42
"In the V1-V2 analysis, the extracted latent space was too big to be visualized. Do the authors think their method in general will find applications in dimensionality reduction / visualization of trial-to-trial variability?"

In most of the previous studies applying dimensionality reduction to neural population activity, the extracted latent space has dimensionality higher than 3 and cannot be directly visualized (e.g., Yu et al., 2009, JNP). In that sense, this study is no different. If one wishes to visualize the latent space, one can orthonormalize the latent dimensions (i.e., columns of C) and visualize just the top dimensions, that explain most of the shared variance. Furthermore, one can take advantage of recently-developed visualization tools, such as DataHigh (Cowley et al., 2013, JNE), to visualize latent spaces of dimensionality greater than 3.

Assigned_Reviewer_42
"Computational considerations? Out of the 3200 available trials, only 1000 were used according to the authors. This is rather unusual practice, so I suspect the dataset was down-sampled to ease the computational burden? What dataset sizes do the authors think can be handled by their model?"

We chose to analyze a subset of the trials for rapid iteration of the analyses. Given that 1000 trials provides a total of 246,000 timepoints (at 5 ms resolution), this provides plenty of data to fit any of the models with 128 observed neurons. At the same time, we could speed up the running time considerably by parallelizing the cross-validation procedure. Fitting the model parameters once is reasonably fast (around 15 minutes for gLARA, on a 2.6 GHz Intel Core i7). The time-consuming part is the cross-validation, to fit the model parameters for every possible combination of latent dimensionalities in each brain area, along with the order of the auto-regressive process.